# Heterologous Expression and Biochemical Characterization of a Novel Lytic Polysaccharide Monooxygenase from *Chitinilyticum aquatile* CSC-1

**DOI:** 10.3390/microorganisms12071381

**Published:** 2024-07-08

**Authors:** Xuezhi Shao, Hongliang Fang, Tao Li, Liyan Yang, Dengfeng Yang, Lixia Pan

**Affiliations:** 1School of Chemistry and Chemical Engineering, Guangxi University, Nanning 530004, China; shaoxuezhi611@163.com (X.S.); 15678995426@163.com (H.F.); 2National Key Laboratory of Non-Food Biomass Energy Technology, Guangxi Key Laboratory of Marine Natural Products and Combinatorial Biosynthesis Chemistry, Guangxi Academy of Sciences, Nanning 530007, China; litao0830@163.com (T.L.); yangliyan@gxas.cn (L.Y.)

**Keywords:** AA10 LPMO, characterization, biomass deconstruction, *Chitinilyticum aquatile*, sustainability

## Abstract

Lytic polysaccharide monooxygenases (LPMOs) are copper-dependent enzymes that catalyze the oxidative cleavage of recalcitrant polysaccharides. There are limited reports on LPMOs capable of concurrently catalyzing the oxidative cleavage of both cellulose and chitin. In this study, we identified and cloned a novel LPMO from the newly isolated bacterium *Chitinilyticum aquatile* CSC-1, designated as *Ca*LPMO10. When using 2, 6-dimethylphenol (2, 6-DMP) as the substrate, *Ca*LPMO10 exhibited optimal activity at 50 °C and pH 8, demonstrating good temperature stability at 30 °C. Even after a 6 h incubation at pH 8 and 30 °C, *Ca*LPMO10 retained approximately 83.03 ± 1.25% residual enzyme activity. Most metal ions were found to enhance the enzyme activity of *Ca*LPMO10, with ascorbic acid identified as the optimal reducing agent. Mass spectrometry analysis indicated that *Ca*LPMO10 displayed oxidative activity towards both chitin and cellulose, identifying it as a C1/C4-oxidized LPMO. *Ca*LPMO10 shows promise as a key enzyme for the efficient utilization of biomass resources in future applications.

## 1. Introduction

In the conventional enzymatic degradation of biomass, a diverse array of glycoside hydrolases, encompassing chitinase, cellulase, and hemicellulase, collaborate effectively to target the crystalline polysaccharides that are naturally abundant in various biomasses [1,2,3]. These enzymes efficiently catalyze the hydrolytic breakdown of these polysaccharides, ultimately yielding oligosaccharides or mono-saccharides as the end products. Therefore, enhancing the yield of glycoside hydrolases and developing feasible combinations to boost subsequent fermentation reactions have emerged as focal points in research. In recent years, the emergence of lytic polysaccharide monooxygenases (LPMOs) has opened up a new way of degrading biomass [4,5,6]. This enzyme is a copper-dependent monooxygenase, with its active site featuring a mononuclear copper ion that chelates with the N-terminal histidine residue [7,8,9]. This unique characteristic, rare in nature, potentially grants LPMOs significant oxidation capabilities [10]. As a result, they hold great promise for improving processes by boosting the activity of enzyme consortia that degrade polysaccharides [11].

According to the auxiliary activity (AA) family in the CAZy (Carbohydrate-Active enZYme Database), LPMOs are presently categorized into eight families: AA9-11 and AA13-17 [12], primarily consisting of genes from bacteria, fungi, unicellular, and multicellular algae, as well as animals [13,14,15]. Among these, AA9 and AA10 are the most extensively researched. However, while AA9 LPMOs are exclusively distributed in fungi and exhibit activity against cellulose or hemicellulose [16,17], AA10 LPMOs are predominantly present in bacteria and demonstrate a more diverse substrate specificity, with chitin activity, cellulose activity, or both [11,17]. Nevertheless, there are currently relatively few reports on enzymes from the AA10 family that exhibit the ability to catalyze the oxidative cleavage of both cellulose and chitin. Notably, *Sc*LPMO10B and *Sc*LPMO10C from *Streptomyces coelicolor* have been shown to be active on both chitin and cellulose [11]. Furthermore, *Ma*LPMO10B and *Ma*LPMO10D from *Micromonospora aurantiaca* have also shown dual-substrate specificity [18,19].

The reaction process of LPMO involves three key elements: a copper ion active center, hydrogen peroxide or molecular oxygen as a co-substrate [20,21], and external electron donors. These electron donors can be provided by substances like ascorbic acid, other low-molecular-weight reducing agents, and certain oxidoreductases [22]. The strong oxidation mechanism of LPMO depends on the redox reaction initiated by its coordinated copper [6] and the availability of oxidizing substances [7]. It was later demonstrated that LPMO, along with the co-substrate H_2_O_2_ [20], cleaved the β-1,4-glycosidic bond in the polysaccharide by oxidizing the C1 or C4 position of the crystalline polysaccharide in the presence of external electrons. This process results in the formation of C1, C4, and C1/C4 oxidation products [21,23]. Among these activities, C1 activity results in the formation of lactones in equilibrium with uronic acid. On the other hand, C4 activity produces 4-aldo-keto-sugar, which then hydrates to gemdiols. Lastly, C1/C4 activity generates uronic acid and 4-aldo-keto-sugar products [24]. It is noteworthy that, among the AA10 family members reported to date, all of them exhibit either C1 oxidation sites or both C1 and C4 oxidation sites. Interestingly, no members with solely C4 oxidation sites have been identified [25]. This observation could be attributed to the level of accessibility of the active copper site in the axial position with the ligand [26]. Among them, the majority of LPMOs in the AA10 family are C1 active, while only a few, such as *Tf*AA10A [27], *Ma*LPMO10B [18], *Sc*LPMO10B [28], and *Tt*AA10A [29], have been reported to possess C1/C4 bi-functional oxidation abilities. Among these, only *Sc*LPMO10B [28] and *Ma*LPMO10B [18] have demonstrated activity on both chitin and cellulose substrates.

In conclusion, the number of reported C1/C4 bi-functional LPMO enzymes with dual-substrate specificity remains limited, underscoring the significant research potential of these enzymes. In this study, the *C. aquatile* CSC-1 strain [30] was isolated from local sediment by our laboratory. Through bioinformatics analysis, its genome was found to contain a unique lytic polysaccharide monooxygenase belonging to the AA10 family. Notably, LPMOs from *Chitinilyticum* sp. have not been previously reported. Phylogenetic tree construction and homologous sequence alignment analysis of *Ca*LPMO10 were performed to determine its family origin. Subsequently, the target gene was cloned into the pET-22b plasmid and successfully heterologously expressed in *Escherichia coli* to obtain a high-purity target active protein. The enzymatic properties of *Ca*LPMO10 were characterized, with mass spectrometry identifying it as a C1/C4-oxidized LPMO capable of acting on both chitin and cellulose.

## 2. Materials and Methods

### 2.1. Bacterial Strains, Plasmid, Chemicals, and Media

*C. aquatile* CSC-1, *E. coli* BL21-CodonPlus (DE_3_)-RIPL (BL21 RIPL), and pET-22b (+) were stored in our laboratory. Phosphoric acid ammonium sulfate, sodium acetate, Tris, K_2_HPO_4_, KH_2_PO_4_, metal compounds (ZnCl_2_, FeCl_3_, MnCl_2_, CoCl_2_, NiCl_2_, CuCl_2_, BaCl_2_, LiCl, and NH_4_Cl), acetone, boric acid, and β-mercaptoethanol were supplied from Guoyao, China; Isopropyl β-D-1-thiogalactopyranoside (IPTG), glycine, hydrochloric acid (HCl), NaOH, potassium sodium tartrate, ethylalcohol, Avicel^®^PH-101(MCC), α-chitin, and 3,5-dinitro salicylic acid (DNS) were supplied from Sigma-Aldrich, Darmstadt, Germany; and Catechol, Hydroquinone, 3, 4-Dihydroxybeenzoic acid, Gallic acid, L-Ascorbic Acid, and Homovanillic acid were supplied from Aladdin, China. Trifluoroacetic acid was sourced from TCI, Japan. 2, 6-Dimethoxyphenol was sourced from Heowns, Tianjin, China. The Ni-Sepharose 6 FF column was from Smart-Lifesciences (Changzhou, China). *Nco* I and *Xho* I restriction enzymes and TriTrack DNA loading dye were supplied from TaKaRa, Osaka, Japan, while Phanta Max Mastr Mix was supplied from Vazyme, Nanjing, China, and Yeast extract, Tryptone was supplied from OXOID, Basingstoke, UK.

### 2.2. Bioinformatics Analysis of CaLPMO10

The *CaLPMO10* gene was predicted to encode amino acids, the molecular weight (Mw), and the isoelectric point (pI) via the Expasy website (Expasy—Compute pI/Mw tool). Domain prediction was performed using InterPro (InterPro (ebi.ac.uk, URL (accessed on 12 May 2023))) analysis and protein domain structure visualization was completed in DOG2.0 (https://dog.biocuckoo.org/, URL (accessed on 12 May 2023)) [31]. Phylogenetic analysis was performed using MEGA-X-10.0.5 software and TVBOT web server (https://chiplot.online/tvbot.html, URL (accessed on 12 May 2023)) [32] was used for display and annotation. The full-length *Ca*LPMO10 and five identified AA10s (*Tm*LPMO10A, *Sc*LPMO10D, *Tf*LPMO10B, *Sam*LPMO10C, and *As*LPMO10B) were aligned using the ClustalX-2.1. Sequence similarity and secondary structures were analyzed by ESPript 3.0 web server (https://espript.ibcp.fr/ESPript/ESPript/, URL (accessed on 12 May 2023)) [33]. The structure of *Ca*LPMO10 was constructed using AlphaFold2 V2.2.4 [34]. The three-dimensional model structure of *Ca*LPMO10 was visualized using PyMOL 2.4 software.

### 2.3. Expression and Purification of the Recombinant CaLPMO10

The total DNA of strain CSC-1, isolated and screened by our laboratory, was extracted by a bacterial DNA extraction kit (TIANGEN, Beijing, China). The *CaLPMO10* gene was amplified by specific primers (Appendix A) and inserted into the pET-22b (+) vector using *Nco* I and *Xho* I as recognition sites. The plasmid was then transformed into *E. coli* BL21 RIPL for protein expression. The positive colony was then cultured in 10 mL of LB medium with 10 μL of 0.1 g/mL Ampicillin at 37 °C overnight. The culture was subsequently transferred to 1 L of LB medium at a final volume ratio of 1% and incubated at 37 °C until the OD_600_ reached 0.6 to 0.8. Following this, a final concentration of 1 mM IPTG was added to induce protein expression. The induction process was conducted at 16 °C for 16 h. Subsequently, the recombinant cell was harvested and resuspended in a lysis buffer containing 500 mM NaCl and 50 mM Tris-HCl buffer (pH 8.0), sonicated for 30min, and the lysate was centrifuged at 4 °C. Following a previously reported protocol [30], the supernatant was purified using Ni-Sepharose 6 FF column. After removing imidazole with a 20 mM pH 8.0 Tris-HCl buffer using a Millipore^®^ ultrafiltration tube, the purified protein was saturated with a three-fold molar excess of CuSO_4_ at 4 °C for 30 min [35]. Subsequently, the excess copper ions were removed using a desalting resin column (PD MidiTrap G-25) in a 20 mM Tris-HCl buffer and concentrated to the desired concentration using an ultrafiltration tube. The enzyme activity was determined using 2, 6-DMP as a substrate [36]. The enzyme activity unit of the *Ca*LPMO10 recombinant enzyme was defined as follows: in a 200 μL reaction system, the amount of enzyme required to convert the substrate to 1 μmol of coerulignone in 1 min at 30 °C was considered one enzyme activity unit (U).

### 2.4. Characterization of Enzymatic Properties of CaLPMO10

#### 2.4.1. Physicochemical Properties of *Ca*LPMO10 Using 2, 6-Dimethoxyphenol

The activity of *Ca*LPMO10 was tested in a 50 mM pH 8.0 Tris-HCl buffer at various temperatures (20–70 °C) with a final concentration of 1 mM 2, 6-DMP, 2 μM *Ca*LPMO10, and 0.1 mM H_2_O_2_. The highest activity was set as 100% activity. The activity was evaluated by monitoring the oxidation of 2, 6-DMP at 469 nm (ε469 = 53,200 M^−1^ cm^−1^). For thermal stability, *Ca*LPMO10 was incubated at 30 and 50 °C for different durations, and residual activity was measured at optimum conditions. The initial activity was set to 100% without any heat incubation.

To study pH effects, *Ca*LPMO10 activity was measured at 30 °C in buffers with pH ranging from 5.0 to 9.0, using citrate–phosphate buffer (pH 5.0–6.5), sodium dihydrogen phosphate–disodium hydrogen phosphate buffer (pH 6.5–8), and glycine-NaOH buffer (pH 8.0–9.0) as a buffer system. Highest activity was set as 100%. For pH stability, *Ca*LPMO10 was incubated in buffers with pH 6–8 for 1 and 24 h, and residual activity was determined. Initial activity was set as 100%.

In order to determine the effect of metal ions on the activity of *Ca*LPMO10, different metal ions were selected to detect the effect on the activity of *Ca*LPMO10: Zn^2+^, Fe^3+^, Cu^2+^, Mn^2+^, Ba^2+^, Ni^2+^, Co^2+^, Li^1+^, and NH_4_^1+^. The metal ions were initially prepared as a 1 M concentration solution, which was then gradually diluted for subsequent use. The enzyme activity determination system (200 μL) comprises the following components: 4 μM *Ca*LPMO10, 50 mM pH 8.0 Tris-HCl, 0.1 mM H_2_O_2_, 1 mM 2, 6-DMP, and a final concentration of 0.1 mM of various types of metal ions. The absorbance change at 469 nm was subsequently measured. The experimental group without metal ions but containing *Ca*LPMO10 was used as the control group (CK). Additionally, considering the unique characteristics of LPMOs reactions, the final activity was subtracted from the activity resulting from the addition of metal ions alone (without *Ca*LPMO10) and from the total enzyme activity when both metal ions and *Ca*LPMO10 were added.

#### 2.4.2. The Effect of Different Reducing Agents on the Activity of *Ca*LPMO10

Various reducing agents were chosen as electronic donors to facilitate the detection of *Ca*LPMO10 activity. These included catechol, hydroquinone, 3,4-dihydroxybeenzoic acid, gallic acid, L-ascorbic acid, and homovanillic acid. The reducing agents were prepared as a 100 mM solution with dimethyl sulfoxide and stored in the dark at 4 °C. The reaction system consisted of 100 μL with 4 μM *Ca*LPMO 10, 1 mM reducing agent, and 100 mM pH 8.0 sodium phosphate buffer. Subsequently, 100 μL Amplex Red working solution was added to the reaction system, and the absorbance change at 560 nm was measured [37]. The H_2_O_2_ produced was calculated based on the H_2_O_2_ standard curve (Appendix A) after the absorbance reached a stable value. The control group did not contain *Ca*LPMO10.

### 2.5. Oxidative Activity Identification of CaLPMO10

To investigate the oxidative activity of *Ca*LPMO10, a 1 mL reaction system was set up with 2 μM *Ca*LPMO10 and 1.0 mM AscA (ascorbic acid). The substrates MCC and α-chitin were present at 2% (*w*/*v*) in a 50 mM pH 8.0 Tris buffer at 30 °C with agitation at 220 rpm for 24 h. Following the reaction, the supernatant was purified and desalted using a graphite carbon solid phase extraction column (Bonna-Agela Technologies, Tianjin, China). Subsequently, matrix-assisted laser desorption/ionization time-of-flight mass spectrometry (MALDI-TOF MS) analysis was carried out as described in Li et al. [38].

### 2.6. Statistical Analysis

All tests were conducted in triplicate, and the data are presented as mean ± standard error (SE). One-way analysis of variance (ANOVA) was performed using Origin software (version 2021).

## 3. Results and Discussion

### 3.1. Amino Acid Sequence Analysis and Structural Model of CaLPMO10

Through sequence analysis, it was determined that *Ca*LPMO10 from *C. aquatile* CSC-1 consists of 511 amino acids. Furthermore, a natural signal sequence of 25 amino acids was predicted using SignalP-5.0. Structural prediction analysis using InterPro revealed that *Ca*LPMO10 contains an AA10 catalytic domain (amino acids 26-204), as well as a GbpA_2 (amino acids 215-317) that is connected to two CBM5/12 domains (amino acids 360-404 and 461-507) (Figure 1A). It has been found that bacterial LPMOs from the AA10 family (PF03067) are active on chitin or/and cellulose substrates [11,28], and have more diversity in terms of carbohydrate modules attached to them. The AA10 family usually contains one or more modules of type A (CBM2, 5, 10, 12, and 73), while type A CBM is composed of a flat substrate-binding surface rich in aromatic residues, which is easier to bind to insoluble and highly crystalline cellulose or chitin [39], but has little affinity for soluble carbohydrates. The CBM5/12 binding domain contained in *Ca*LPMO10 is more conducive to the binding of refractory crystalline polysaccharides.

A phylogenetic analysis of the *Ca*LPMO10^CD^ (catalytic domain) was conducted by comparing it with identified or reported LPMO10 catalytic domains using MEGA7.0 software. Additionally, a landscaping mapping was performed using the TVBOT online website for further analysis and visualization. The phylogenetic analysis of *Ca*LPMO10 in relation to characterized or reported LPMO10s [40,41] revealed its placement within clade II, indicating its close relationship to LPMO10s known for their cellulose activity (Figure 1B). The results from the multiple sequence alignment indicated that the catalytic domain of *Ca*LPMO10 (*Ca*LPMO10^CD^) contained two characteristic Histidines (His 1 and His 103) involved in copper binding at the active site, as well as a conserved aromatic Phenylalanine residue (Phe174) (Figure 1C). Sequence alignment analysis showed that the *Ca*LPMO10^CD^ exhibited low homology with other reported AA10 family LPMO enzymes. Among the characterized LPMOs of the AA10 family, *Ca*LPMO10^CD^ exhibited the highest sequence identity to *Sc*LPMO10D^CD^ [42] and *Tf*LPMO10B^CD^ [43] at 48.09% and 41.97%, respectively. Additionally, *Ca*LPMO10^CD^ showed significant sequence identity with other enzymes in the AA10 family, including 37.34% with the atypical chitin-active LPMO *Cj*LPMO10A^CD^ [40] and 35.05% with the well-known studied cellulose-active *Sc*LPMO10C^CD^ (CelS2) [44]. Combined with sequence alignment analysis, the L2 loop (Tyr39—Asn46) motif of *Ca*LPMO10 is closer to the LPMO of the cellulose active motif ((W)NWFGVL) [45].

The predicted three-dimensional structure of *Ca*LPMO10^CD^ shows that, similar to other members of the AA10 protein family, *Ca*LPMO10^CD^ has a core immunoglobulin-like β-sandwich topology. The active site is composed of His1 and His103, which are located in the β-sandwich structure and coordinate copper with Phe174. It is worth noting that, as shown in Figure 2A, Phe174 occupies the axial position of copper coordination and may be involved in substrate binding. The red ring in the diagram is the L2 loop (Tyr39—Asn46) of *Ca*LPMO10, and its terminal amino acid Asn46 is closer to the catalytic center Cu connected by His1, and its side chain is perpendicular to the substrate binding surface and points to the substrate (Figure 2B). This is similar to typical C1/C4 oxidized LPMOs (*Tf*LPMO10A and *Sc*LPMO10B) [46]. These phenomena indicate that *Ca*LPMO10 may be the LPMO of cellulose activity C1/C4.

### 3.2. Expression and Purification of CaLPMO10

The LPMO gene *CaLPMO10* was successfully cloned from *C. aquatile* CSC-1 and expressed as active protein in *E. coli* BL21 RIPL. The recombinant *Ca*LPMO10 was purified using the nickel column affinity method. SDS-PAGE analysis showed a single protein band with a molecular weight of approximately 54.29 kDa, consistent with the predicted molecular weight of the *Ca*LPMO10 protein (Figure 3). The purified recombinant *Ca*LPMO10 demonstrated a purity of nearly 90% (Figure 3), reaching a purification fold of up to 113.82 with a yield of 57.8 (Table 1).

### 3.3. Characterization of CaLPMO10 Using 2, 6-Dimethoxyphenol as Substrate

#### 3.3.1. The Effect of Temperature on the Activity of *Ca*LPMO10

Next, the enzymatic properties of *Ca*LPMO10 were characterized in terms of temperature, pH, and metal ions. The highest oxidation activity was observed at 50 °C, indicating that it is active at mesophilic temperatures. Furthermore, *Ca*LPMO10 displayed good catalytic performance at 40–70 °C (Figure 4A), consistent with previous studies on AA10 LPMOs, which have shown that the optimal temperature range is between 40 and 60 °C [47]. For instance, *Na*LPMO10A exhibited the highest oxidation activity at 40 °C [48]. As depicted in Figure 4B, the thermal stability of *Ca*LPMO10 was evident at varying temperatures (30 and 50 °C). It is observed that, with prolonged incubation time, the stability of *Ca*LPMO10 diminished. After 4h incubation at 30 °C, the residual activity was maintained at approximately 51%.

#### 3.3.2. The Effect of pH on the Activity of *Ca*LPMO10

*Ca*LPMO10 exhibited the highest oxidation activity at pH 8.0 (Figure 4C). Furthermore, its relative activity notably declined under acidic or alkaline pH conditions, suggesting that it displayed optimal activity under weakly alkaline pH conditions. The optimal pH value of *Ca*LPMO10 was found to be similar to that of *Bat*LPMO10 in *Bacillus subtilis* and the majority of the previously reported AA9 LPMOs [49]. The residual enzyme activity of *Ca*LPMO10 exhibited a general decline across the pH range of 6 to 9, corresponding to 20.2%, 68.7%, 83.9%, and 45.0% of the initial activity, respectively (Figure 4D). It can be seen that the enzyme has the highest stability at pH 8. The above studies show that *Ca*LPMO10 has optimal activity and stability under weakly alkaline pH conditions.

#### 3.3.3. The Effect of Metal Ion on the Activity of *Ca*LPMO10

In order to minimize the interference from other ions in the experiment, chloride ions (Cl^−^) were chosen as the anions, as depicted in Figure 5A. Low concentrations of metal ions (final concentration of 0.1 mM), Mn^2+^, Ni^2+^, Co^2+^, and Cu^2+^, have improved the activity of *Ca*LPMO10, which increased by 728.93%, 1198.98%, 2689.81%, and 3139.81%, respectively. Secondly, NH_4_^1+^, Li^1+^, and Ba^2+^ could also slightly increase the activity of *Ca*LPMO10, which were 121.15%, 136.92%, and 164.76%, respectively. However, when Zn^2+^ and Fe^3+^ are added, *Ca*LPMO10 quickly deactivates and produces a large amount of protein precipitation. It can be seen from the metal ion experiment that different metal ions have a great influence on the enzyme activity of *Ca*LPMO10, especially the promotion effect of copper ions, which also proves that copper ions are the most effective active site metal ions of *Ca*LPMO10, which is consistent with the research results of Stepnov et al. [50].

### 3.4. The Effect of a Reducing Agent on the Activity of CaLPMO10

The Amplex^TM^ Red detection method [51] can be employed to measure the quantity of H_2_O_2_ produced by *Ca*LPMO10 and co-substrate O_2_ in the presence of a reducing agent. Figure 5B illustrates that various reducing agents exhibit distinct effects on the activity of *Ca*LPMO10. When ascorbic acid and gallic acid are utilized as reducing agents, the H_2_O_2_ production is measured at 2.3 μmol and 1.6 μmol, respectively. This notably boosts the activity of *Ca*LPMO10, aligning with previous research findings [48,52]. Ascorbic acid is frequently included in reactions as an electron donor to facilitate the LPMO reaction [22,52]. Additionally, there are reports on the use of other reducing agents like gallic acid and glutathione as electron donors in similar studies [22].

### 3.5. Oxidation Activity Assay of CaLPMO10

To evaluate the oxidation activity of *Ca*LPMO10 on cellulose and chitin, MALDI-TOF mass spectrometry was employed to detect the reaction products. This analysis led to the identification of typical cellulose oxidation products and chitin oxidation products, as depicted in Figure 6. When cellulose served as the substrate (as shown in Figure 6A) and Appendix A, the products primarily appeared as oxidized oligosaccharides concentrated around cellopentasaccharide acid (Glc_4_GlcA), cellohexasaccharide acid (Glc_5_GlcA), and celloheptasaccharide acid (Glc_6_GlcA). For instance, the C4 oxidation product was Glc_4_GlcA + Na (*m*/*z*: 849.31); C1 oxidation products included Glc_4_GlcA-H + 2Na (*m*/*z*: 889.27), Glc_5_GlcA + Na (*m*/*z*: 1029.83), Glc_5_GlcA-H + 2Na (*m*/*z*: 1051.98), and Glc_6_GlcA-H + 2Na (*m*/*z*: 1213.68); and C1, C4, and C6 oxidation products included Glc_4_GlcA + 2Na (*m*/*z*: 879.27) and Glc_6_GlcA + 2Na (*m*/*z*: 1203.71). When chitin was employed as the substrate (as depicted in Figure 6B), the products were predominantly found in the peaks of oxidized oligosaccharides corresponding to (GlcNAc)_2_GlcNAcA, (GlcNAc)_3_GlcNAcA, (GlcNAc)_4_GlcNAcA, and (GlcNAc)_5_GlcNAcA. The characteristic chitin double oxidation product peaks included (GlcNAc)_2_GlcNAcA-H + 2Na (*m*/*z*: 688.25), (GlcNAc)_3_GlcNAcA-H + 2Na (*m*/*z*: 891.29), and (GlcNAc)_5_GlcNAcA-H + 2Na (*m*/*z*: 1297.45). The products observed in the mass spectrometry analysis align with earlier findings on LPMO10’s cellulose and chitin oxidation activity [53,54]. This suggests that *Ca*LPMO10 functions in the C1/C4 oxidation process when acting upon cellulose and chitin substrates.

## 4. Conclusions

The *CaLPMO10* gene was obtained from the *C. aquatile* CSC-1 strain and heterologously expressed in *E. coli*. The molecular weight of the purified recombinant *Ca*LPMO10 protein was about 54.29 kDa. The effects of temperature, pH, and metal ions on the activity of the enzyme were studied with 2, 6-DMP as the substrate. The optimum temperature and pH were determined, and the relative activity of the enzyme was maintained in a wide range of pHs and temperatures. In addition, it has good temperature and pH stability. Mn^2+^, Ni^2+^, Co^2+^, Cu^2+^, and other metal ions have a promoting effect on the activity of *Ca*LPMO10, and Cu^2+^ has the greatest promoting effect. Most of the reducing agents enhanced its activity, with ascorbic acid being the most suitable reducing agent. Mass spectrometry was used to identify the oxidation products of *Ca*LPMO10 on cellulose and chitin, with the detection of typical uronic acid and 4-aldo-keto-sugar indicating the oxidation activity of *Ca*LPMO10 on cellulose (MCC) and chitin (α-chitin). *Ca*LPMO10 was classified as a C1/C4 LPMO. Given these characteristics, *Ca*LPMO10 shows significant potential for a broad range of applications in both basic research and industrial settings in the future.

## Figures and Tables

**Figure 1 microorganisms-12-01381-f001:**
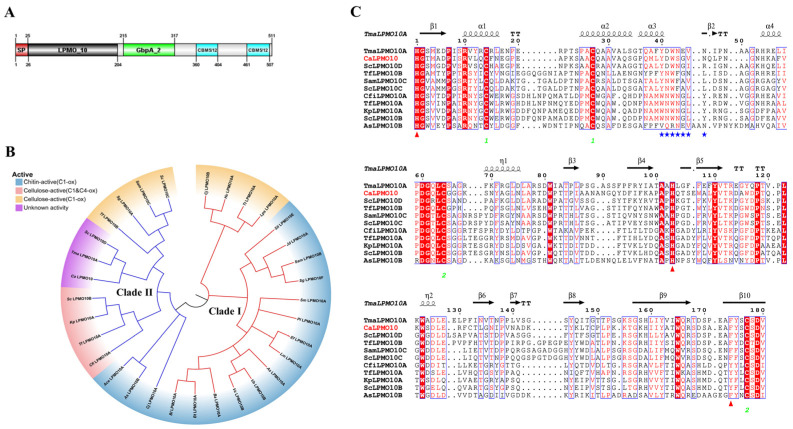
Bioinformatics analysis of *Ca*LPMO10. (**A**). Domain analysis of *Ca*LPMO10. (**B**). Phylogenic tree of *Ca*LPMO10 and other annotated AA10s shows clustering in clade I (red) and clade II (blue), with labels indicating AA10s having distinct substrate activity. (**C**). Sequence alignment of *Ca*LPMO10 with other identified AA10s. *Ca*LPMO10 is represented by red, with the active site being indicated by a red triangle and the blue asterisk symbolizing the L2 ring. The identical amino acid sites have been highlighted against a red background. Green numbers represent disulfidebonds.

**Figure 2 microorganisms-12-01381-f002:**
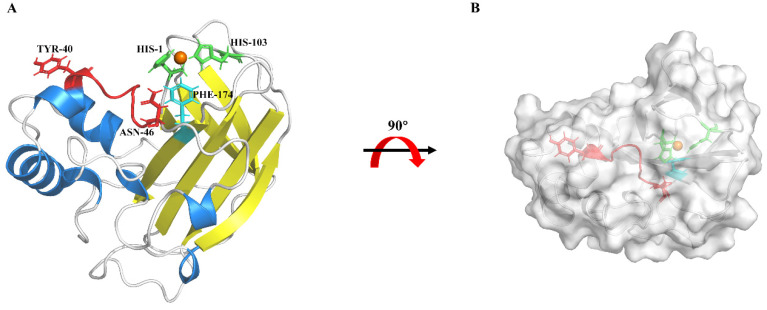
Prediction of the three-dimensional structure model of *Ca*LPMO10. For (**A**), the active sites of histidine (His1 and 103, marked green) and phenylalanine (Phe174,marked blue) bind to the protein surface with the substrate, and the active site copper ion is displayed on the gold ball. The red region represents the L2 ring, with key amino acids tyrosine (Tyr40) and asparagine (Asn46). (**B**) is obtained by rotating the left image of the *Ca*LPMO10 three-dimensional model structure outward by 90°.

**Figure 3 microorganisms-12-01381-f003:**
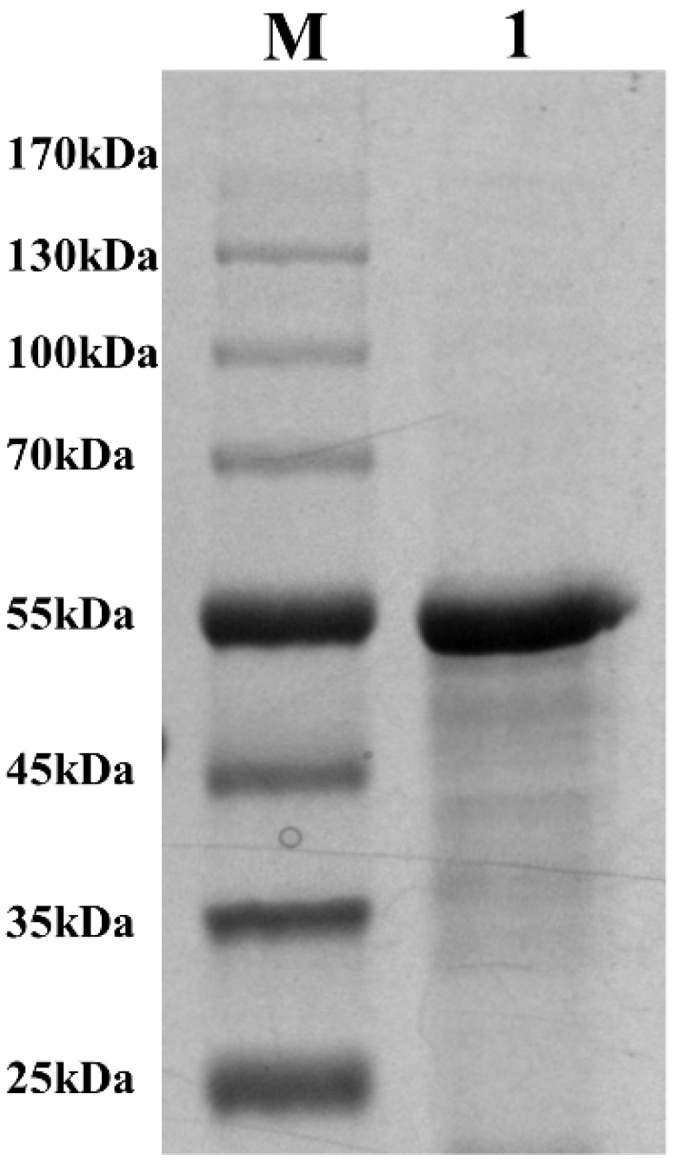
SDS-PAGE detection of *Ca*LPMO10. Lane M: broad range protein marker; Lane 1: the purified protein eluent.

**Figure 4 microorganisms-12-01381-f004:**
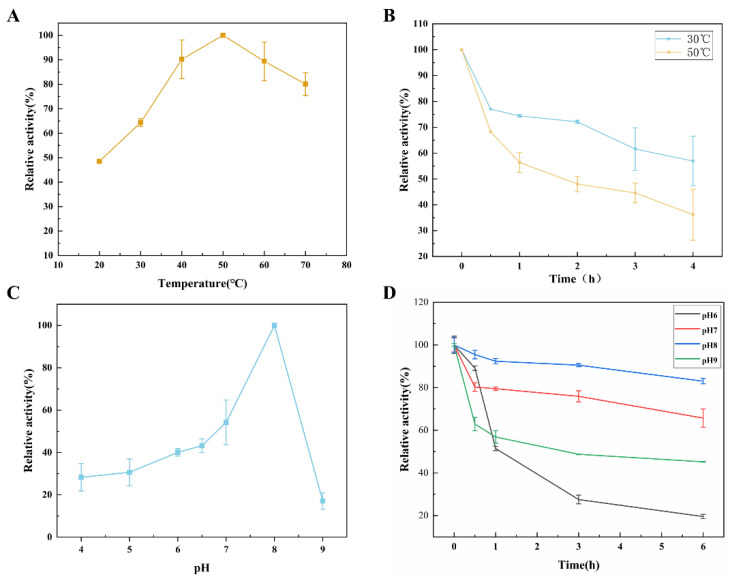
Characterization of the purified recombinant *Ca*LPMO10. (**A**) The optimum temperature of *Ca*LPMO10. (**B**) The thermostability of *Ca*LPMO10. (**C**) The optimum pH of *Ca*LPMO10. (**D**) The pH stability of *Ca*LPMO10.

**Figure 5 microorganisms-12-01381-f005:**
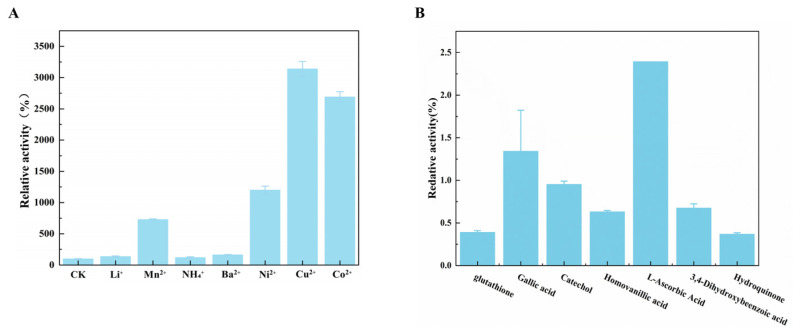
The effect of metal ions (**A**) and a reducing agent (**B**) on the activity of *Ca*LPMO10.

**Figure 6 microorganisms-12-01381-f006:**
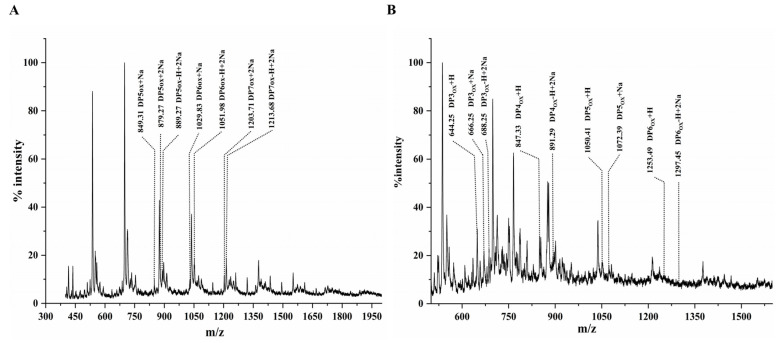
Mass spectrum of the production of cellulose and chitin by *Ca*LPMO10 with ascorbic acid as the electron donor. (**A**): Mass spectra of products of *Ca*LPMO10 acting on cellulose; (**B**): mass spectra of products of *Ca*LPMO10 acting on chitin.

**Table 1 microorganisms-12-01381-t001:** Purification table of recombinant *Ca*LPMO10.

Purification Step	Volume (mL)	Total Protein (mg)	Activity (mU/mL)	Total Activity (mU)	Specific Activity (mU/mg)	% Yield	Purification Fold
Crude extract	30	1739.28	0.224	4.484	0.00025779	100	1
Ni-NTA chromatography	15	8.84	0.173	2.594	0.293437	57.8	113.82

## Data Availability

The original contributions presented in the study are included in the article/Appendix A, further inquiries can be directed to the corresponding authors.

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
