# Peer review of "Heterologous Expression and Biochemical Characterization of a Novel Lytic Polysaccharide Monooxygenase from Chitinilyticum aquatile CSC-1"

_microorganisms, 2024, doi:10.3390/microorganisms12071381_

Round 1

Reviewer 1 Report

Comments and Suggestions for Authors

This biochemical study of the lytic polysaccharide monooxygenase (LPMO) of Chitinilyticum aquatile CSC-1 enhances our understanding of the catalytic properties of the LPMO family enzymes and contributes to various industrial applications using recalcitrant crystalline polysaccharides.

Major points

1. Since there are many studies on LPMOs, the overall significance of this study should be more clear. To understand why this CaLMPO10 was chosen for this study, details about Chitinilyticum aquatile CSC-1 and CaLPMO10 should be explained in the Introduction.

2. How was CaLPMO10 selected? Does the bacterium have any other LPMOs besides CaLPMO10?

3. The authors stated that the molecular weight of the recombinant protein is estimated to be 54.29 kDa from SDS-PAGE, but the molecular weight markers (Figure 3) clearly show that the molecular weight of this protein is smaller than 50 kDa. In the conclusion section, the authors stated the molecular weight of the enzyme is 50-51 kDa. Those undermine the reliability of the results. Please correct them adequately.

4. Several major peaks are not identified in Figure 6. The manuscript states that the degradation product peaks are dominant, but there are many other peaks that are not assigned.

From these spectra alone, it is very difficult to determine if the enzyme is really acting on these polysaccharides. Furthermore, it is also difficult to determine the mode of degradation.

Without analyzing changes of the spectra over time, it is impossible to assert from these noisy spectra and state that these polysaccharides are degrading by C1/C4 oxidations.

At the very least, the authors should compare the spectrum before and after the enzymatic reaction to clarify the peak of the degradation products.

Others

In the manuscript, there are too many minor mistakes. It is impossible to point out everything. It makes peer review difficult. Please revise well.

1. Abstract, L7: At what temperature was the enzyme stable?

2. P2, L49, “metal ions (ZnCl2, FeCl3…)”: these molecules are not ions.

3. P3, L5, “EcoRV”: NcoI and XhoI appear in later sentences, but not EcoRV.

4. P3, L27, “0.1 g/mL Ampicillin”: 0.1 g/L?

5. P3, L33, “Following a previous reported protocol”: What part of the method do the authors refer to? And which are the references?

6. P3, L42, “wood vinegar”: Is it true?

7. P4, L7, “pH 6-8 for 1and 24h”: At what temperature did the authors experiment?

8. P5, last line, Figure 2 legend, etc.: Which is correct, Phe173 or Phe174?

Reviewer 2 Report

Comments and Suggestions for Authors

Microorganisms

The manuscript “Heterologous expression and biochemical characterization of a novel lytic polysaccharide monooxygenase from Chitinilyticum aquatile CSC-1” by Shao et al., describes the expression and characterization of a C1/C4 oxidizing LPMO. The manuscript is scientifically sound and can be of interest to the readership of “Microorganisms”. However, some major issues must be resolved before publication:

Page 5: “Additionally, CaLPMO10CD showed significant sequence homology with other en- zymes in the AA10 family, including 37.34% homology with the atypical chitin-active LPMO CjLPMO10ACD [43], and 35.05% homology with the well-known studied cellulose- active ScLPMO10CCD (CelS2) [47]” Proteins are homologous or are not, there exist no %homology.

Page 6: “We conducted a characterization of the relationship between the enzymatic properties of CaLPMO10 and temperature and pH”. Please improve the text.

Figure 4 (D): A scatter plot should be used instead of bars.

Page 8: Effect of metal ions: how were the controls performed? How do the authors explain 30-fold activity? (comparison with literature?)

Page 8: “However, when Zn2+ and Fe3+ are added, CaLPMO10 will quickly deactivate and produce a large amount of protein precipitation.” Please avoid the present/future form.

Figure 6: Are there any controls?

All over the text: please avoid subjective statements such as “EXCELLENT catalytic performance”, “BEST activity and stability”, “have GREATLY improved”, “NOTABLY boosts the activity”, “the H2O2 production is RELATIVELY SUBSTANTIAL”, “metal ions have a SIGNIFICANT promoting effect”. Moreover, typing errors should be corrected.

Please do not hesitate to contact me if you have any inquires.

Sincerely yours

Comments on the Quality of English Language

Typing errors should be corrected.

Round 2

Reviewer 1 Report

Comments and Suggestions for Authors

The manuscript has been well revised according to the reviewers’ suggestions. I recommend this manuscript for publication.

Author Response

Thank you very much.

Reviewer 2 Report

Comments and Suggestions for Authors

This new version was improved according to the reviewers suggestions. A minor modification is suggested: the x-axis of figure 4D should represent the real distance in relationship with the time (the distance between 0 and 0.5 h cannot be the same as the distance between 1 and 2 h). 
